# Study of the Dissolution of Stainless-Steel Slag Minerals in Different Acid Environments to Promote Their Use for the Treatment of Acidic Wastewaters

**Mattia De Colle** [1,*], **Ross Kielman** [1], **Andreas Karlsson** [2], **Andrey Karasev** [1] **and Pär G. Jönsson** [1]

1 KTH Royal Institute of Technology, SE-100 44 Stockholm, Sweden; ross.kielman@live.com (R.K.); karasev@kth.se (A.K.); parj@kth.se (P.G.J.)
2 Swedish Museum of Natural History, SE-104 05 Stockholm, Sweden; andreas.karlsson@nrm.se
* Correspondence: mattiadc@kth.se

**Abstract:** Several stainless-steel slags have been successfully employed in previous studies as substitutes for lime in the treatment of industrial acidic wastewaters. This study deepens the knowledge of such application, by analyzing the neutralizing capacity of different slags related to their mineral compositions. To do so, firstly the chemical and mineral compositions of all the slag samples are assessed. Then, 0.5 g, 1 g, 2 g of each slag and 0.25 g and 0.5 g of lime are used to neutralize 100 g of 0.1 M HCl or $HNO_3$ solutions. After the has neutralization occurred, the solid residues are extracted and analyzed using XRD spectroscopy. Then, the solubility of the minerals is assessed and ranked, by comparing the XRD spectra of the residues with the obtained pH values. The results show that minerals such as dicalcium silicate and bredigite are highly soluble in the selected experimental conditions, while minerals such as merwinite and åkermanite, only partially. Moreover, Al-rich slags seem to perform poorly due to the formation of hydroxides, which generate extra protons. However, when the weight of slag is adequately adjusted, Al-rich slags can increase the pH values to higher levels compared to the other studied slags.

**Keywords:** steelmaking slag; solubility; minerals; wastewater treatment; leaching

## 1. Introduction

In 2019, 52 Mt of stainless-steel has been produced worldwide [1], which generated roughly 15 to 17 Mt of slag [2]. Compared to other kinds of by-products, such as Basic Oxygen Furnace (BOF) or Blast Furnace (BF) slags, which are well recycled in state-of-the-art applications, stainless-steel slags are mostly disposed in landfills. As an example, in Sweden only, stainless-steel slags are roughly 20% of the total amount of slags produced during the steelmaking processes. However, they constitute around 70% of the landfilled output [3]. The high concentrations of leachable cancerogenic elements (such as Cr or Ni) in these slags, impedes their use in the common state-of-the-art applications, namely aggregates for road pavements or cement. Therefore, these kinds of slags are mostly disposed in landfills. Landfilling itself constitutes a problem too, because the same phenomenon generates risks of soil and water stream contamination, if not properly mended [4–8]. In addition, the production of stainless-steel has been steadily growing since the year 2008 [1], resulting in higher volumes of landfilled stainless-steel slags. Thus, it is imperative to find a solution for these kinds of materials, as landfills will progressively get filled. Furthermore, environmental restrictions are projected to tighten in the nearby future, resulting in an increasingly high economic liability for stainless steel producers.

One possible solution to the problem is to develop a novel application for these kinds of products. Several studies have already tested mineral by-products for the neutralization and treatment of industrial waste waters [9–14]. These waste streams can substitute the use of raw materials such as quicklime, which is the standard product used for water

treatment, due to the high solubility and alkalinity of CaO. Steel and stainless-steel slags contain high percentages of CaO and MgO, which make them suitable replacements to quicklime. Previous studies [15,16] demonstrated that several stainless-steel slags could be employed for the neutralization and treatment of waste waters derived by the pickling process in steelmaking. The waste waters contained a dilution of different acids like HF, HCl, HNO$_3$ and H$_2$SO$_4$, which also presented high levels of dissolved metallic elements. Previous experiments demonstrated that as well as lime, the stainless-steel slags tested could rise the required pH values during the waste waters neutralization processes, while guaranteeing a comparable removal of metallic ions from the liquid phases [15]. However, more systematic investigations are necessary to evaluate the efficiency of various stainless-steel slags in neutralizing the pH of acidic solutions. In fact, it is seen from previous studies [10,16] that the mineral composition of different slags affects the slags capacity of buffering acidic waters.

Mineral dissolution and element leaching are the phenomenon that dictates the reaction of slags and acidic waste waters. The dissolution of mineral phases in aqueous media is a complex phenomenon that varies vastly depending on the various parameters of the environment in which the solid materials are in contact with. The phenomenon is usually studied in terms of leaching behavior of slags, which is a detrimental effect when the material is used in standard applications as cement or asphalt mixtures. In the body of literature analyzed for this study, the pH value of the solution is considered one of the most important factors in the dissolution of the mineral phases. Experimentally, it was demonstrated that the more alkali in the solution, the lower the dissolution of the minerals [17,18]. This is also in compliance with more theoretical studies that used geochemical models to determine the leaching of mineral phases present in steel slags [19]. In batch tests where the pH is not kept constant, the dissolution of the mineral phases is limited by the increase of the solution's pH value. This phenomenon is well reported by De Windt et al. [19]. In fact, the authors found that by reducing the L/S (liquid to solid) ratio by a factor of 10, it only increases the pH values from 11.2 to only 11.9. This means that at alkaline pH values, using 10 times the mass increases the pH value by less than a unit value. Similar results were obtained by Mombelli et al. [20], which showed that only at L/S = 100 L/kg the slag is significantly dissolved. Contrary, for L/S = 10 L/kg the reactions between liquid and solid phases are only cortical, meaning that the dissolution of the mineral phase is stifled substantially by the alkalinity of the solution. Different dissolution rates at different pH levels are also reported by several other studies [17,18]. Although, when the pH value is continuously adjusted and kept to low values, the dissolution of minerals progresses to completion [9,10].

The pH value is not the only parameter that affects the dissolution of mineral phases. Several studies report different ion mobilities depending on the phases being dissolved. De Windt et al. [19] shows how Ca and Si are the most leached elements from BOF slags, due to the high solubility of larnite. On the other hand, Mg and Al show very low mobilities since they are associated with Fe-bearing phases with very low solubilities. However, Cunha et al. [9] report high solubility levels of Mg and Al, as well as a varying dissolution of Si depending on the material tested. Moreover, Mombelli et al. [20] show an increased dissolution of Mg when the amount of dissolved Ca is low, while the dissolved Al content seems to increase the more Ca is being dissolved. Therefore, there seems to be no consensus regarding an independent leaching behavior of single elements. In fact, the aforementioned studies affirm that the dissolution of the mineral phases determines the solubilities of the elements leached in the solution [9,19,20]. Overall, all the studies agree that Ca is the most mobile element, and most Ca-bearing minerals found in metallurgical slags are at least partially soluble in water.

In addition, different dissolved elements modify the pH value in different ways [17,18]. The theoretical chemical equilibria of the most common and abundant elements in steelmaking slags are listed in the following equations:

$$CaO(s) + 2H^+ \rightarrow Ca^{2+} + H_2O \tag{1}$$

$$MgO(s) + 2H^+ \rightarrow Mg^{2+} + H_2O \tag{2}$$

$$Al_2O_3(s) + 6H^+ \rightarrow 2Al^{3+} + 3H_2O \tag{3}$$

$$Al^{3+} + 3H_2O \rightarrow Al(OH)_3(s) + 3H^+ \tag{4}$$

$$Al(OH)_3(s) + H_2O \rightarrow Al(OH)_4^- + H^+ \tag{5}$$

$$SiO_2(s) + 2H_2O \rightarrow Si(OH)_4(aq) \tag{6}$$

$$Si(OH)_4(aq) \rightarrow SiO(OH)_3^- + H^+ \tag{7}$$

As is it possible to notice from Equations (1) and (2), Ca and Mg present the same chemistry: the oxides dissociate in a bivalent proton and a water molecule. This reaction raises the pH value of the solution by removing dissolved protons. However, the chemistry of Al and Si is more complex. Al oxides dissolves by consuming 6 protons forming 2 trivalent Al ions and 3 water molecules. In addition, Al is involved in two more reactions which both generate a proton as shown in Equations (4) and (5). Si also forms a proton when the hydrated form of its oxide dissociates in aqueous media. Pourbaix diagrams have been used in previous studies to determine the favored species as a function of the pH value [17]. The diagrams are calculated using FactSage 6.1, according to a study made by Bale at al. [21].By analyzing the graphs is it possible to notice that silic acid does not dissociate until very high levels of pH. Therefore, it does not contribute negatively to the increase of pH of the solution. On the contrary, $Al^{3+}$ is the favored species only at very low pH values. Afterwards the dominant species becomes $Al(OH)_3$ (s), which forms a proton as the result of its formation. Therefore, Al is expected to counteract the rise of the pH values caused by other elements such as Ca and Mg.

The present study aims at continuing the investigations on stainless-steel slags and their abilities to buffer the pH values of acidic waters, by improving the methodology developed in previous experiments [15,16]. The following experiments are designed to determine the effect of different mineral compositions on the pH buffering capacities of slags. The same kinetic conditions used in previous studies are maintained, as they also affect the final pH value obtained [15,16]. Furthermore, this study also tests the pH buffering capacities of slags in different acidic environments, as they have only been tested against specific kinds of acidic waste waters.

## 2. Materials and Methods

### 2.1. Materials Preparation and Characterization

The current study utilizes the same slag samples used by the authors in previous experiments [15,16]. The samples are taken by two different Swedish steel plants, namely Outokumpu Stainless AB (Avesta, Sweden) and Sandvik Materials Technology (Sandviken, Sweden). Two slag samples from each company were taken from their respective landfill sites (O1 and S1). Moreover, one slag sample was taken directly from the Electric Arc Furnace (EAF) process (O2) and one from the Argon Oxygen Decarburization (AOD) converter process (S2). The samples were ball milled and sifted by using a 25–50 μm mesh, to obtain a well-defined size interval of slag particles to be used for the experimental trials.

To assess the mineral composition of the slag samples, identification and semi-quantitative analysis of the mineral phases within each slag sample was performed using scanning electron microscopy (SEM) and powder X-ray powder diffraction (XRD). Energy spectra for distinct mineral phases were obtained using an FEI Quanta 650 field-emission scanning electron microscope (FEG-SEM) equipped with an 80 mm$^2$ X-Max$^N$ Oxford Instruments energy-dispersive spectroscopy (EDS) detector operating with an accelerating voltage of 20 kV. The spectra were reduced to chemical data within the AZtec software and subsequently calculated to approximate mineral formulas. Thereafter, semi-quantification of the slag minerals in each sample was achieved using a PANalytical X'Pert$^3$ Powder diffractometer equipped with an X'celerator silicon-strip detector and operated at 40 mA and 45 kV (CuK$\alpha$-radiation, $\lambda = 1.5406$ Å). Peak positions were determined using the X'Pert

HighScore Plus 4.6 program (peak positions were corrected against an external Si Metal Standard, NBS640b) and matched with mineral phases identified during SEM analyses.

To also have a precise quantitative assessment of the elements present in the slag samples (such as Ca, Si, Al, Mg among others), the chemical compositions of all four slag samples have been identified using Sector Field Inductively Coupled Mass Spectrometry (ICP-SFMS). The samples have been digested in $HNO_3$, HCl and HF prior to ICP-SFMS to extract all the metal ions present. Thereby, the concentrations of 32 metallic elements were determined.

### 2.2. pH Buffering Trials

After assessing both the mineral and chemical compositions of the slag samples, the focus shifted towards testing a reliable method that could be used to compare each sample with respect to its neutralization capacity. All slag samples were used as reactants for the neutralization of different acidic solutions of known pH values. Two HCl and $HNO_3$ 0.1 molar (M) solutions were prepared, by mixing commercial grade chemicals (Hydrochloric acid 1 M and a solution of 65 wt% Nitric Acid) with distilled water. In every neutralization trial, 100 g of acidic solution were weighted and used as a standard quantity to be neutralized. The quantity chosen for measuring the amount of the solution was its weight, rather than its volume. This is because the former can be measured more precisely than the latter in laboratory conditions. The molar concentration of the acidic solution, corresponding to its pH levels, was chosen in accordance with the pH levels of common industrial acidic waste waters used in previous experiments [15,16]. Since the pH values of the acidic waste waters varied in a range between 1 and 2, a pH value of 1 was chosen for the HCl and $HNO_3$ solutions. The experimental procedure was also replicated from previous experiments [15,16]. The trials were conducted by using a VWR pHenomenal IS 2100 L a pH-meter mounting a sensor with Pt wire junction, and a liquid electrolyte (Phenomenal LS 221). Before each trial, the pH meter was calibrated by using three standard buffers of known pH values of 4, 7 and 10. The neutralization method selected for the experimental trials was performed as follows:

1. The initial pH value of the acidic solution was measured ($pH_0$).
2. The beaker with 100 g of the acidic solution was placed on a magnetic stirrer, a magnet was placed inside the beaker, and the stirring started using a speed of 480 rpm.
3. All the reactant was dropped into the solution at time t = 0.
4. After that, the pH was measured at intervals of 10 min, for 90 min in total ($pH_{10}$, $pH_{20}$, ..., $pH_{90}$). To do so, 30 s before the measurement the stirring was stopped so the suspension could precipitate at the bottom of the beaker. After 30 s, the pH electrode was inserted in the beaker and the pH level was measured. The value was determined once a stable reading was obtained, or after 1 min from the insertion of the electrode.
5. After the reading was taken, the pH electrode was removed from the beaker. Successively, the stirring started again with a speed of 480 rpm.
6. The trial was stopped after 90 min. Thereafter, the beaker was sealed with PARAFILM® M until drying occurred.

In total 18 trials with this method were conducted using different slag types, slag weights and acidic environments depending on the scope of the investigation. A first tranche of trials tested each slag three times, by using different weights of slag. During these 12 trials a 0.1 M HCl solution was used. To facilitate the comparison between different slags and highlight the effect of composition, for the current study three fixed weight values of 0.5, 1 and 2 g of slag were used. The only exception was applied to slag type S2, where the trial using 0.5 g additions was replaced with a replication of the trial using a 2 g addition of slag. A second tranche of trials tested additions of 1 g of each slag into a 0.1 M $HNO_3$ solution. The last tranche of two trials used a standard grade CaO powder as a reactant. Specifically, additions of 0.25 g and 0.5 g of CaO were tested against a 0.1 M HCl solution. The weight values of lime were chosen by performing simple chemical

calculations. The reactions between the acids and CaO, ideally follow Equations (8) and (9). Since the stoichiometry is identical for both reactions, it is possible to derive that 0.28 g of ideal CaO is needed to rise the pH value of an ideal 0.1 M solution of monoprotic acid (HCl or $HNO_3$) up to exactly 7.

$$CaO\ (s) + 2HCl\ (aq) \rightarrow CaCl_2\ (aq) + H_2O\ (l) \tag{8}$$

$$CaO\ (s) + 2HNO_3\ (aq) \rightarrow Ca(NO_3)_2\ (aq) + H_2O\ (l) \tag{9}$$

### 2.3. Extraction and Characterization of the Reaction Products

After the neutralizations, all the samples were subjected to the same treatment for the extraction and investigations of the reaction products. The beakers were placed in a ventilated oven at 90 °C overnight, so the water and eventually the excess of HCl or $HNO_3$ could evaporate. Once the evaporation phase was completed, the dried samples were scraped from the beakers and ground in a mortar. Afterward, the powders were placed on a Petri dish and dried again in a ventilated oven at 105° for 30 min to eliminate all remaining moisture. Thereafter, the samples were once again ground with a mortar and prepared for the XRD analysis. The analyses were performed using a 2θ angle ranging from 5° to 70° with an increment of 0.01°/step and an acquisition time of 1 s/step (model Bruker D8 DISCOVER, equipped with a Cu K-α radiation source at 40 kV and 40 mA). After analyzing the results of the XRD analysis, in case of slag sample S2 SEM-EDS analyses (model Hitachi S-3700N) were used to estimate the compositions of the different phases present in the reaction products. In preparation of the SEM-EDS analysis, the powders were mounted on an epoxy resin and polished to a mirror finish. Specifically, the samples were ground with a 1200 grit disk and polished with a pad of 3 μm and 1 μm sprayed with a diamond suspension of the respective particle size. Afterwards, the samples were coated with Au to improve the image quality during the SEM-EDS investigations.

### 3. Results

### 3.1. Mineral and Chemical Composition

The results of the semi-quantitative analyses indicate the mineral phases are present in the slag samples. The complete list of the mineral phases found in the samples, along with their chemical compositions and crystal structures, are shown in Table 1. In addition, the distribution of the mineral phases is shown in Figure 1.

**Table 1.** Compound name, chemical formula and crystal system of each mineral found in the slag samples.

| Compound Name | Chemical Formula | Crystal System |
|---|---|---|
| dicalcium silicate γ | Ca2 O4 Si1 | Orthorhombic |
| bredigite (O1) | Ca26.93 Ba0.59 Mg3.62 Mn0.86 Si16.00 O64.00 | Orthorhombic |
| bredigite (S1) | Ba0.3 Ca13.5 Mg1.8 Mn0.4 O32 Si9 | Orthorhombic |
| fluorite | Ca1 F2 | Cubic |
| magnesiochromite (O1) | Mg6.96 Fe1.04 Cr16.00 O32.00 | Cubic |
| magnesiochromite (O2) | Cr2 Mg1 O4 | Tetragonal |
| magnesiochromite (S1) | Al7.78 Fe3.59 Mg4.70 Mn0.05 Si0.01 Zn0.05 Cr7.78 Ni0.01 Ti0.02 O32.00 | Cubic |
| åkermanite (O1) | Ca4.00 Mg1.42 Al1.02 Si3.48 O14.00 | Tetragonal |
| åkermanite (S1) | Ca4.00 Mg0.92 Al1.98 Si3.04 O14.00 | Tetragonal |
| cuspidine | Ca16.00 Si8.00 O28.00 F8.00 | Monoclinic |
| merwinite | Ca3 Mg1 O8 Si2 | Monoclinic |
| dolomite | Ca3.00 Mg3.00 C6.00 O18.00 | Hexagonal |
| periclase | Mg1 O1 | Cubic |
| portlandite | H2 Ca1 O2 | Hexagonal |
| magnesioferrite | Mg8.00 Fe16.00 O32.00 | Cubic |
| mayenite | Al14 Ca12 O33 | Cubic |

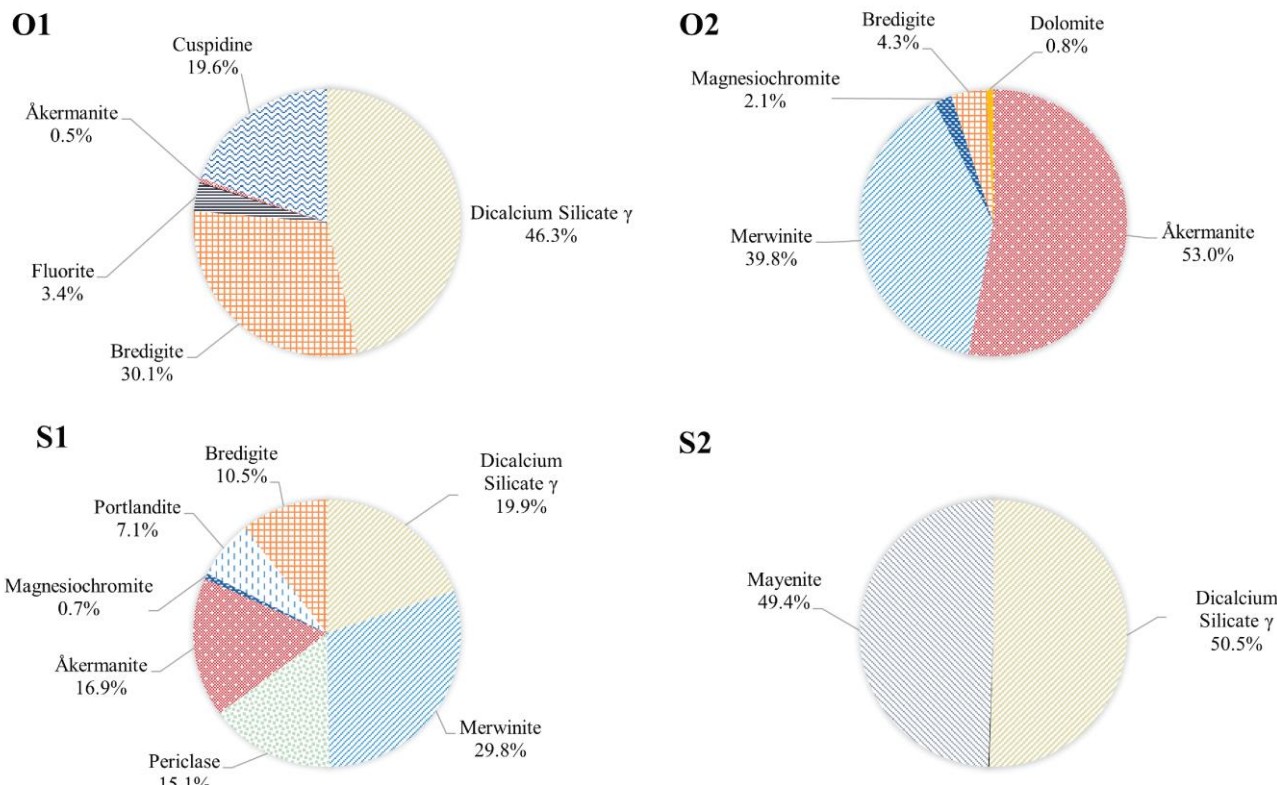

**Figure 1.** Frequencies of the mineral phases, expressed as % of the total, of the four slag samples.

As can be seen in Figure 1 the slag sample S1 contains the highest number of minerals, since it contains 6 phases with a percentage higher than 5%. Sample O1 follows, where 3 phases constitute approximately 96% of the sample content. Sample O2 is almost a binary system consisting of åkermanite and merwinite, as they constitute approximately 93% of the total content. Slag sample S2 is instead a binary system consisting of mayenite and dicalcium silicate γ. The complex multi-mineral landfill slags (O1 and S1) seemed to have a more varied structures than the ones taken from AOD (S2) and the EAF (O2) samples. This is not surprising, as the landfill slags are usually made up of a mix consisting of several by-products and they also have been subjected to additional processes while being exposed to the outdoor environment. dicalcium silicate γ, åkermanite and bredigite are the most common phases, which are present in at least 3 samples. This is in accordance with the results of several studies that reported high amounts of dicalcium silicate γ and bredigite in several kinds of slags [22–24]. Another important factor to mention is that besides magnesiochromite and periclase, all the minerals are Ca-bearing. Therefore, Ca resides in several different mineral structures, which likely present different solubilities and reactivities depending on the acid environments and crystal structures, as shown in previous studies that analyzed the dissolution of minerals often present in metallurgical slags [17,18].

The bulk chemical composition of all the slag samples have been assessed using ICP-SFMS. The analysis measured the concentrations of 32 metallic elements present in the slag samples. The concentrations of the tested elements are shown in Table 2. The most abundant ones in each slag (among the one tested) are also grouped in Figure 2. Overall, Ca is the most abundant element (among the one tested) in all slag samples, where the content ranges from 27% to 32%. Si is the second most abundant element in slag samples O1, O2 and S1. The content ranges from 10% to 14%, while in slag sample S2 it is only 3%. Mg is the third most abundant element for all samples, where the content ranges between 3% and 7%. Al is the fourth most abundant element in slag samples O1 and O2 and the fifth highest for slag sample S1 with a content ranging from 2% to 3%. In case of slag sample S2,

Al is the second main constituent having a 12% weight content, due to the high amount of mayenite. For slag sample S1, the fourth most abundant element is Fe. In addition, the fifth element in weight percentage is Fe for slag sample O1, Mn for slag samples O2, and Cr for slag sample S2 but all with percentages below 1%.

**Table 2.** Amount, expressed in mg/kg, of 32 metallic elements present in the slag samples.

| Element (mg/kg) | O1 | O2 | S1 | S2 |
|---|---|---|---|---|
| Al | 33,100 | 24,200 | 21,800 | 120,000 |
| As | 0 | 6.48 | 3.65 | <3 |
| Ba | 74.9 | 688 | 190 | 60.9 |
| Be | <0.5 | 0.508 | <0.5 | <0.5 |
| Ca | 319,000 | 274,000 | 273,000 | 320,000 |
| Cd | 0.66 | 0.101 | <0.06 | <0.05 |
| Co | 14.4 | 4.26 | 6.73 | 0.653 |
| Cr | 2090 | 3900 | 4870 | 4130 |
| Cu | 40.7 | 24 | 26.7 | <1 |
| Fe | 6650 | 5080 | 32,100 | 1120 |
| Hg | <0.05 | <0.05 | <0.05 | <0.05 |
| K | 511 | 1140 | 1090 | <100 |
| Mg | 41,200 | 48,200 | 70,000 | 37,800 |
| Mn | 2060 | 12,000 | 8940 | 724 |
| Mo | 224 | 72.8 | 221 | 22 |
| Na | 429 | 1180 | 785 | <100 |
| Nb | 73.8 | 505 | 554 | 26.1 |
| Ni | 828 | 203 | 567 | 48.1 |
| P | 86.7 | <50 | 233 | <50 |
| Pb | 14.2 | 18.5 | 3.39 | 4.3 |
| S | 1790 | 1200 | 1700 | 2740 |
| Sb | 0.709 | 0.459 | 0.528 | 0.117 |
| Sc | 1.3 | 1.18 | 1.12 | 1.14 |
| Si | 106,000 | 142,000 | 106,000 | 30,400 |
| Sn | 2.38 | 1.99 | 1.66 | 0.526 |
| Sr | 176 | 232 | 183 | 207 |
| Ti | 2490 | 9730 | 2960 | 565 |
| V | 44.8 | 131 | 175 | 93.3 |
| W | 5.84 | 3.12 | 19 | 1.65 |
| Y | 5.79 | 7.39 | 8.35 | 4.21 |
| Zn | 177 | 9.84 | 79.6 | <4 |
| Zr | 108 | 300 | 145 | 67.9 |

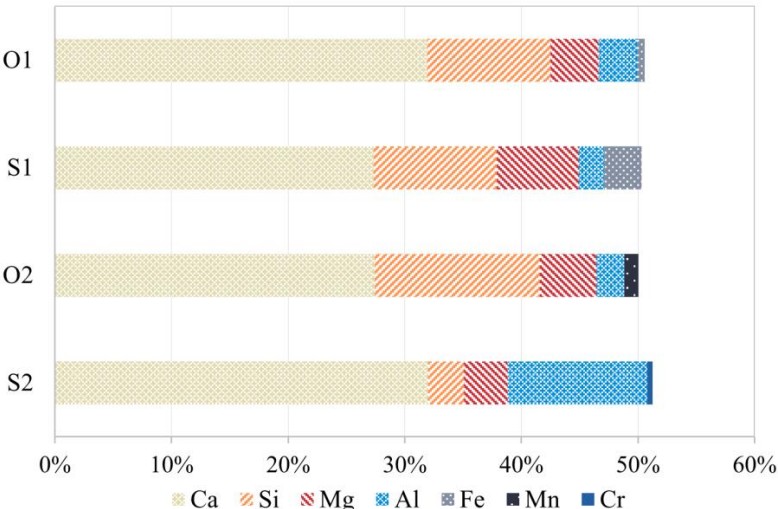

**Figure 2.** Concentrations (in wt%) of the most abundant elements of all the slag samples.

### 3.2. Effect of Weight and Composition of Added Slag

The first twelve experiments were aimed at understanding the relationship between the weight of the added slag and its neutralization capacity. For such trials, a 0.1 M HCl solution was used. The results of the trials are grouped by slag type and the results are shown in Figure 3. For three slag samples beside slag S2, a 1 g addition was sufficient to neutralize the acidic content of the solution and to buffer the pH to values > 7 during a sixty-minutes experiment. Moreover, for all the slags, an addition of 2 g buffered the pH value to the final levels higher than the trials performed with 1 g. The reaction rate during the first 10 min is also faster when 2 g additions are used. This is noticeable, for example, by observing the differences in pH values measured at the beginning and the end of the trials. In fact, the gap in pH levels between the 1 g and 2 g addition curves is the widest at the beginning of the trials, but then it shrinks as the curves start to converge. Specifically, the pH levels obtained with slag sample S1 10 min after the start of the reaction are 5.2 and 8.8 for the curves with 1 g and 2 g addition, respectively. On the other hand, the pH level after 90 min is 8.1 when a 1 g addition is used and 9.5 with a 2 g addition. Thus, at the beginning of the trial the pH values are 64% and 93% of the respective final values. The same happens when slag sample O1 is used: 75% of the final pH value is obtained when a 1 g addition is used, while 91% of the final value is reached when a 2 g addition is used in the first 10 min of the trials. For slag sample O2, the same percentages are 59% and 89%. Doubling the weight of slag increased the final pH value by approximately 0.9 in case of slag O1, 1.5 for slag S1 and 0.78 for slag O2. Finally, the trials using 0.5 g additions confirmed that the quantity chosen is insufficient to neutralize the acidic solution up to a pH value of 7. In fact, the minimum quantities of the given O1, O2 and S1 slags needed to reach pH values > 7 during 90 min, are between 0.5 and 1 g per 100 g of 0.1 M of HCl solution.

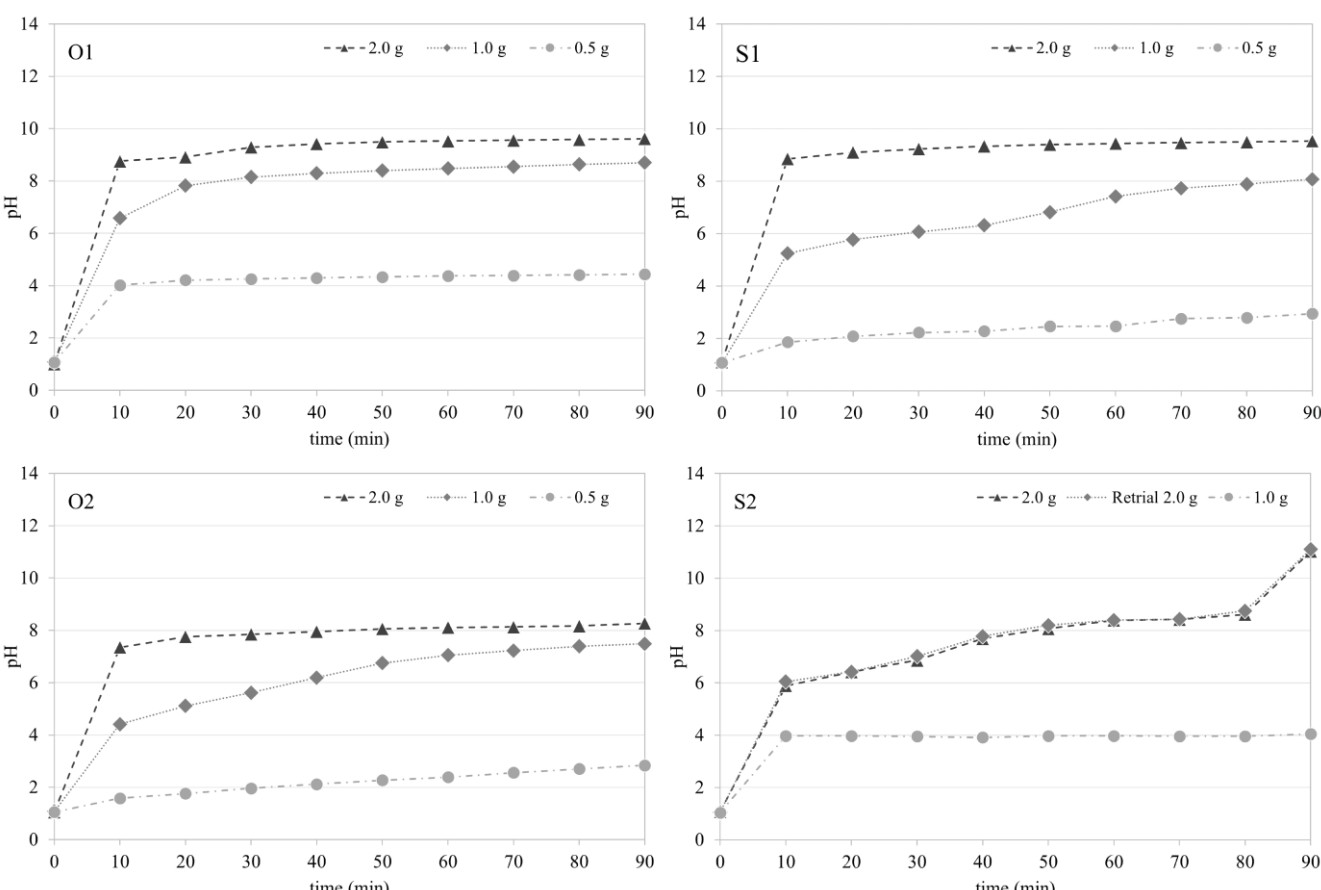

**Figure 3.** Neutralization trials of a 0.1 M HCl solution performed with different weights of the given slag samples.

Slag sample S2 represents an interesting outlier that is worth examining as a separate case. In fact, despite its Ca content is comparable to other slag samples, especially to slag sample O1, the trial conducted with a 1 g addition of slag only reached a pH value of 4. This is in contrast with the results for the other three slag samples used. When 2 g of the same material was tested, the reaction profile up to minute 80 resembled the results for the other curves obtained with different reactants. Although, after minute 80 a sudden rise of the pH value up to 11, was detected. A retrial with the same quantity was carried out that confirmed the anomaly. In case of sample S2, the trial with 0.5 g was not performed. This is because the trial with 1 g could not provide a complete neutralization of the acid.

The slag samples were also grouped based on the trial weight in Figure 4. For this analysis, slag S2 was excluded given its different behavior. Furthermore, the trials using 0.5 g additions since they do not provide a complete neutralization of the acidic content. When a 1 g addition is used, slag O1 is visibly the best reactant: it reaches higher levels of pH and faster compared to the other two slags. When a 2 g addition is used, there is no visible difference between the O1 and S1 samples, which show overlapping curves. However, slag sample O2 reaches a final pH level of approximately pH 1.3 lower than the other two slags (pH 9.6, 9.5 and 8.2 for samples O1, S1 and O2 respectively).

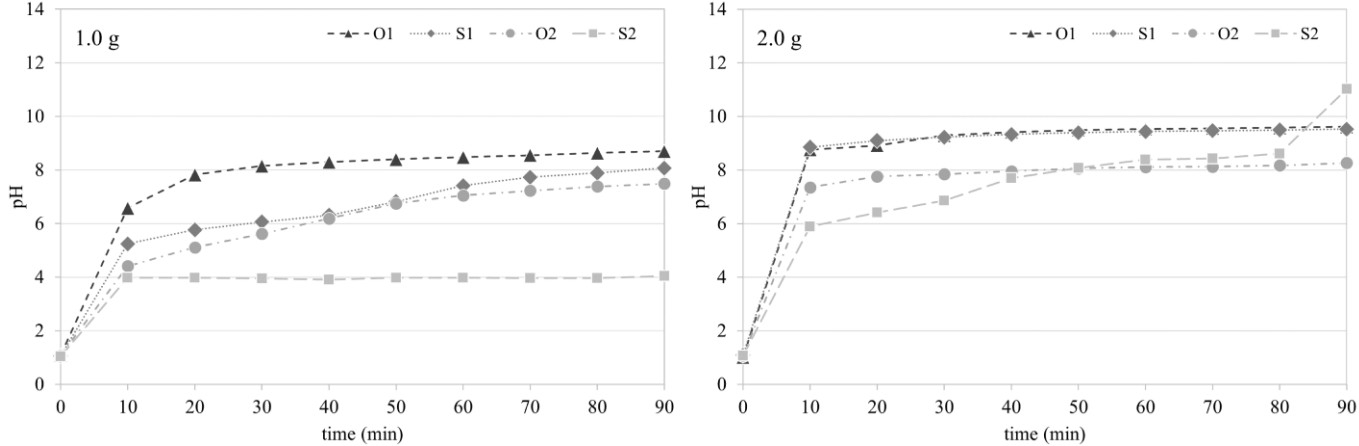

**Figure 4.** Neutralization trials of a 0.1 M HCl solution performed with 1 and 2 g of the slag samples S1, O1 and O2.

Lastly, in the final tranche of the experiments, which used pure CaO as a reactant, the final pH value obtained during 90 min by using 0.25 g was approximately 1.8, and the solution presented no particulate (i.e., unreacted CaO). It was expected that the trial using a 0.25 g addition of CaO will not be able to completely neutralize the acid content, given the ideal chemical calculations performed before the trials. On the other hand, the trial using a 0.5 g addition buffered the acidic solutions to a pH value of 12.5 during the first 10 min, while some remaining CaO was present as a suspension in the solution. This is in line with the reported saturation limit of CaO [25]. The results of the CaO trials are shown in Figure 5.

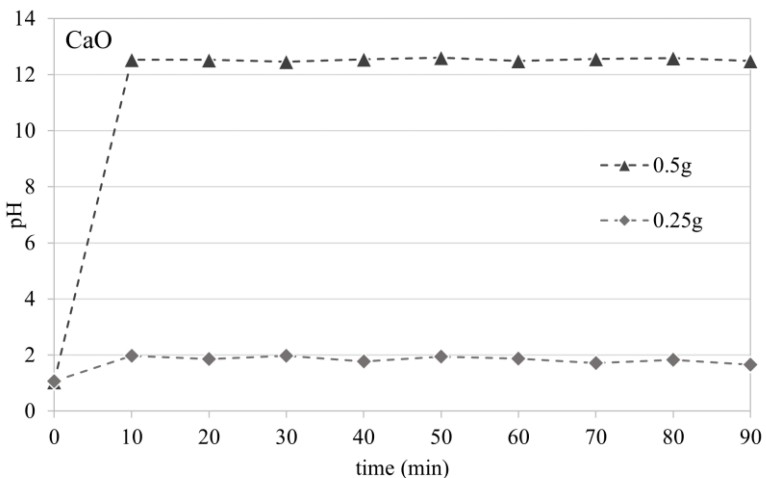

**Figure 5.** Neutralization trials of a O.1 M solution of HCl performed with 0.5 g and 0.25 g of pure CaO.

### 3.3. Effect of Acid Environment

The trials using 1 g addition were replicated also using a solution of $HNO_3$ with the same molarity as the HCl solution (0.1 M). This set of trials was performed to verify whether the slag samples had different solubilities in different acid environments, translating to differences between obtained pH levels. As it is possible to notice from Figure 6, where all the slags were grouped based on their type and acid environment, no meaningful variation in pH level was detected. In fact, the differences in measured pH values in both acidic solutions were between 0.05 and 0.1 for slag samples O1, S1 and S2. Only for slag O2 there was a slight difference of 0.7 between the two final pH values. Regarding the remaining slags, the curves of the test performed with $HNO_3$ overlap almost perfectly with the HCl ones.

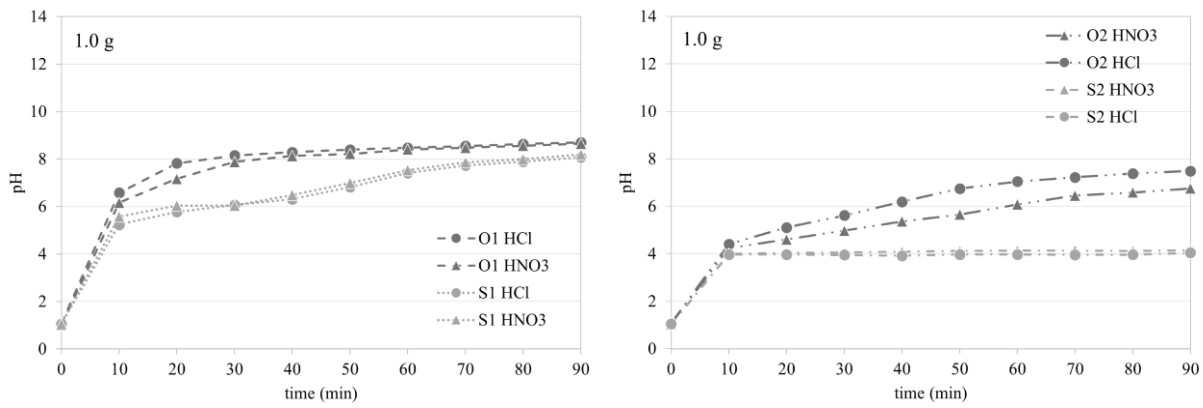

**Figure 6.** Neutralization trials of the 0.1 M HCl and $HNO_3$ acid solutions performed with 1 g of slag samples S1 and O1 (**left** and O2 and S2 (**right**).

### 3.4. XRD Analyses

After the neutralization trials had been performed, all the sediments were extracted from the beakers as explained in Section 2.3. The solid residues obtained when using such a method were analyzed using XRD to analyze their composition. Only the trial derived by the usage of a combination of CaO and $HNO_3$ could not provide analyzable sediments. In fact, the residues precipitated as a hydrated salt that formed a paste rather than a powder.

The XRD spectra of the two trials using CaO are shown in Figure 7. All the main peaks found in the trials using a 0.25 g addition of CaO, are identified in the 0.5 g trial too. In the trial using a 0.25 g addition of CaO, the analysis of the spectrum yielded a partial match

with sinjarite, whose chemical formula is $CaCl_2 \cdot 2(H_2O)$. Therefore, $CaCl_2$ is believed to precipitate during the evaporation phase of the liquid phase in a multiple hydrated form having chemical formula $CaCl_2 \cdot nH_2O$. This is in compliance with the results obtained in previous study, which used sulfuric acid instead, precipitating $CaSO_4$ in several hydrated forms [9] In the 0.5 g trial, there are also peaks belonging to at least another phase.

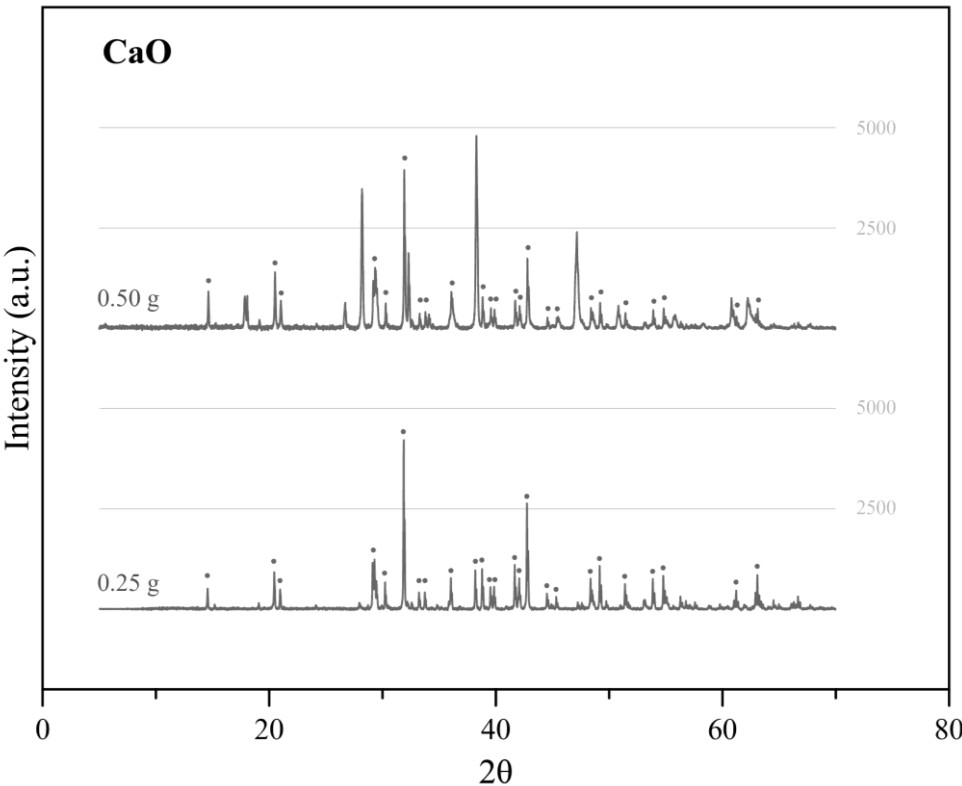

**Figure 7.** XRD spectra of the residues extracted after the neutralization trials of a 0.1 M HCl solution performed with 0.25 g and 0.5 g of CaO. The peaks related with the phase $CaCl_2 \cdot nH_2O$ are highlighted by dots at their respective peaks.

Also, the reaction products obtained during the neutralization trials performed with slag and HCl were extracted from the beakers to determine their compositions. The 11 samples were analyzed using XRD and the spectra are showed in Figure 8. The spectra were grouped based on slag type, to determine the differences in compositions. In all four cases, all the spectra of the reaction products were different when a different quantity of reactant was used. In fact, when the XRD spectra produced by trials with the same slag, but different weights of slag are compared to each other, the peaks positions are the same. However, both relative and absolute intensities are different. This means that the phases in the samples are the same, but their ratio differs, when a different weight of slag is used.

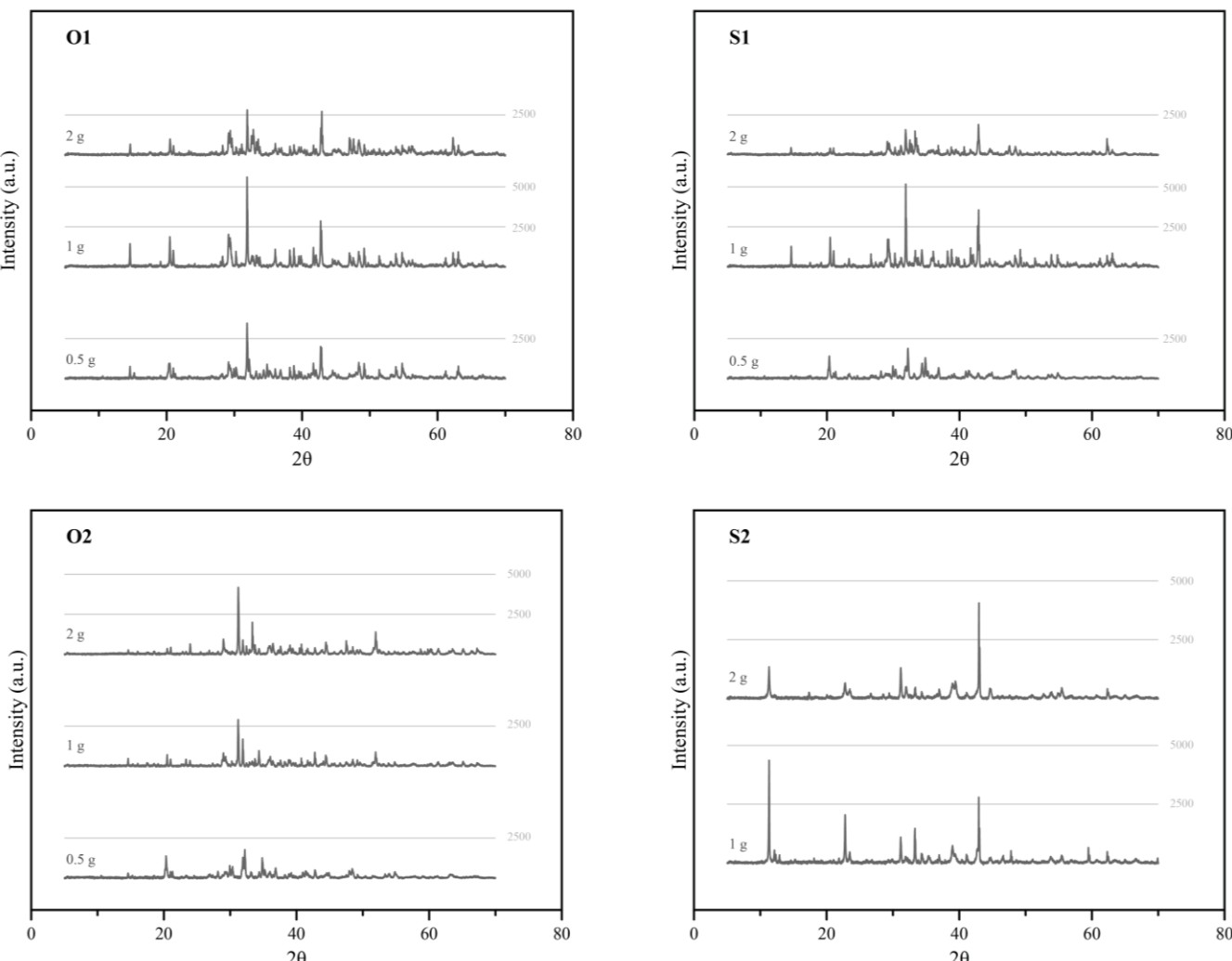

**Figure 8.** XRD spectra of the residues extracted after the neutralization trials of a 0.1 M HCl solution performed with 0.5 g, 1.0 g and 2.0 g of the slag samples.

There are some similarities across spectra produced by different slags. In fact, the results thst were obtained for the trials performed with additions of 1 g of O1, 1 g of S1 and 1 g of O2 seems comparable. Moreover, the spectra produced by those trials are also similar to the one obtained after the trial using a 0.25 g addition of CaO. To facilitate the comparison, the intensities of these 4 spectra have been normalized to a 100% value, as shown in Figure 9. As is it possible to notice, all the main peaks of reaction products obtained after the neutralization made using 0.25 g additions of CaO are present in the spectra of the residues obtained with the slag samples. For slag samples O1 and S1 almost the entire spectra can be described by the $CaCl_2 \cdot nH_2O$ peaks. On the other hand, when looking at the spectrum obtained by the reaction products made by slag O2, the match is only partial, and at least another phase in present. In addition, the peaks corresponding to $CaCl_2 \cdot nH_2O$ seem to decrease in intensity in favor of other phases, either when the initial weight of slag used for neutralization was reduced or increased. It should be pointed out that Slag S2 was not grouped with the others, because the XRD spectra produced (as shown in Figure 8) were clearly different from the other reaction products obtained.

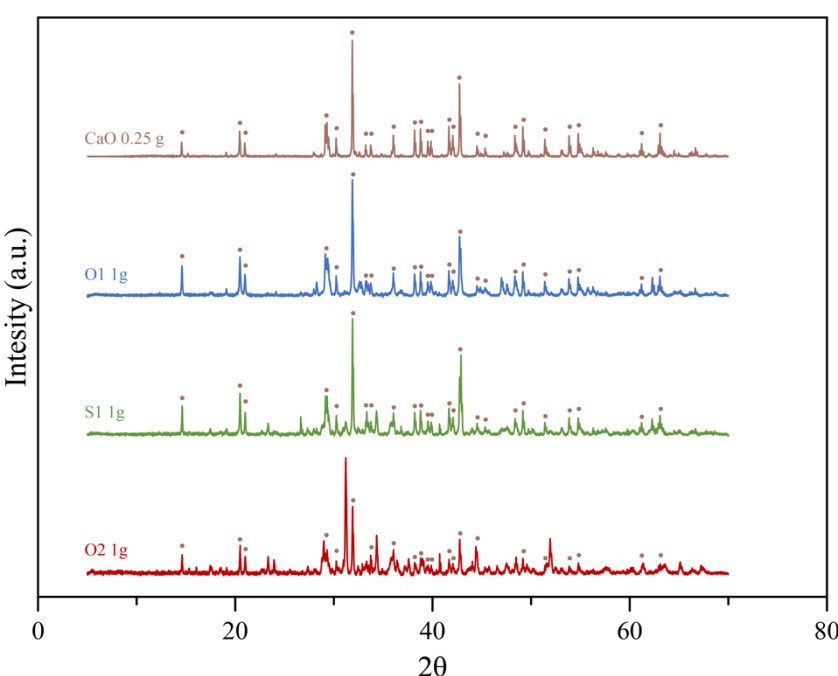

**Figure 9.** XRD spectra of the residues extracted after the neutralization trials of a 0.1 M HCl solution per-formed with 0.25 g of CaO and 1 g of slag samples S1, O1 and O2. The peaks related with the phase $CaCl_2 \cdot nH_2O$ are highlighted by dots at their respective peaks.

## 4. Discussion

The purpose of this research work is to progress the studies on the use of slag as a reactant for acidic waste waters treatment. This application is of interest especially in cases when the slag composition does not easily allow for a proper recycling in the state-of-the-art applications, namely as part of cement or asphalt mixtures. The same kinetic conditions of previous experiments [15,16] has been maintained, although the methodology has been improved in order to better study the effect of different mineral compositions on the neutralization performance. The same four slag samples used in the current research were first tested in a first feasibility study [15]. This previous study did not focus on the particle sizes and mineral compositions of the slag samples. In addition, industrial waste waters with unknown compositions with pH values ranging between 1 to 2 were used. Moreover, the study aimed at measuring the amount of slag needed to reach pH $9 \pm 0.2$ by minute 30 by using a method that is highly influenced by the kinetic of the mineral dissolution. The obtained results showed that only slag samples O1 and S1 properly rose the pH to the desired values, and 40 to 50 g of slag per liter of industrial waste waters were needed. The following study by the same authors reduced and homogenized the particle size ($\leq 63$ μm) of the slag samples and replicated the same experimental methodology [16]. Contrary to previous findings, in that study even slag O2 (EAF slag) reached the desired pH value, but not slag S2 (AOD slag). The study found that the O1 and S1 slag (landfill slags) samples were the best performing ones, utilizing approximately 20–25 g/L to reach a pH value of 9 in 30 min, whereas slag O2 needed roughly 40 g/L to reach the same target. Thus, the quantity of slag needed was halved due to a decrease in particle size. It should be noted that no possible connections between the pH levels obtained and the mineral composition could be drawn, as the latter was still not assessed.

The purpose of the given study is to analyze the differences in mineral compositions between the slag samples and to identify suitable slags for the neutralization of various acidic solutions. To do so, acidic solutions of known composition and controlled pH levels were preferred to industrial acidic waste waters used in previous experiments. Also, the slag particle size distribution was further reduced to diameters between 25 and 50 μm. Moreover, rather than aiming at the same pH levels between trials, the same weight

values of acidic solutions and slag samples were used in the experimental trials, as it was deemed to be more controllable parameters. Therefore, a reliable comparison between pH levels obtained in this and previous study [15,16] was not possible to obtain. Nonetheless, although the current study did not aim at obtaining a pH value of 9.0 ± 0.2 at minute 30, in the trials in the present research performed with 2 g additions (20 g/L) of slag O1 and S1 the pH levels reached were 9.3 and 9.2 at the 30th minute, respectively. When 2 g of slag O2 were added, the pH value was 7.8 at minute 30. These results seem to be in line with the ones found in a previous study [16], where around 20–25 g/L of slag samples O1 and S1 were needed to reach the desired pH values. Also, this is an additional confirmation that slag O2 performs worse than the other studied samples, due to its mineral composition. Regarding the slag sample S2, no quantitative comparison with previous experiments can be done [15,16], since successful neutralizations were never performed with such material before.

Given the complexity of the system involved, each slag samples should be evaluated separately analyzing its initial composition, the XRD patterns of the residues extracted after the neutralization trials, and the final pH levels obtained.

### 4.1. Dissolution of Slag O1

Slag O1 (landfill slag) is the best performing slag sample among the ones tested. When 0.5 g and 1 g are added, the highest final pH values are reached (4.4 and 8.7, respectively). Also, the pH rising rate is the highest among the studied slags (Figure 4). When 2 g additions of slag are used it ties with the results using slag S1 both in terms of the rate of pH increase (Figure 4) and its final values (9.6 and 9.5, respectively). Slag O1 contains the highest amount of Ca (31.9%) and Mg (4.1%) combined, which are the elements considered responsible for increasing the alkalinity of the acidic solutions. The slag is formed mostly by Dicalcium silicate γ ($2CaO \cdot SiO_2$-γ, 46.3% of the total sample composition), bredigite (30.1%) and cuspidine (19.6%). The results of the XRD analysis of the residues, obtained by the neutralization trials performed using a 1 g addition of slag O1, show that all major peaks are attributed to $CaCl_2 \cdot nH_2O$, as shown in Figure 9. The absence of intensity peaks corresponding to the original minerals, suggesting that the original mineral structure is completely disrupted by the end of the trial. Dicalcium silicate γ is a very common phase in metallurgical slags and has been proven to be highly soluble in water [17,20]. High dissolutions rates of cuspidine, especially for acidic pH values, have been observed by He and Suito [26]. No information about the dissolution behavior of bredigite is available in the literature, but the results obtained suggest that the mineral is completely dissolved during the trials. In addition, previous experiments made by Cunha et al. [9], where $H_2SO_4$ was used to dissolve several oxidic products, showed that the residues were formed by $CaSO_4$ as the main constituent, precipitated in several hydrated forms.

### 4.2. Dissolution of Slag O2

Slag O2 (EAF slag) is the third best performing slag based on the final pH levels obtained per unit of weight, despite the fact that it has the lowest total of Ca (27.4%) and Mg (4.8%) contents. When 1 g is used, the slag barely surpasses neutral pH values. Furthermore, when a 2 g addition is used, the final pH values are comparable with the pH values obtained using 1 g additions of slags S1 and O1. In fact, the final pH values reached with 1 g additions of slags O1 and S1 are 8.7 and 8.1 respectively. When a 2 g addition of slag O2 is added, a final pH value of 8.3 is obtained. Contrary to the composition obtained by using slags O1 and S1, $CaCl_2 \cdot nH_2O$ is not the most abundant phase present in residues obtained when using slag O2. In fact, as shown in Figure 9, the peaks associated with $CaCl_2 \cdot nH_2O$ only partially describe the XRD pattern of the residues obtained in the neutralization trial. Also, according to Figure 8 the abundance of the phase decreases in the 2 g trials in favor of the other phases. This is a common observation for slags S1 and O1 because the amount of $Cl^{2+}$ ions dissolved is the same in both trials, as the concentration of the acidic solution is the same. Thus, when the evaporation of the liquid phase occurs,

the same amount of CaCl$_2$·nH$_2$O is formed when 1 g and 2 g additions are used. By using 1 g more gram of slag, the CaCl$_2$·nH$_2$O ratio diminishes. At the same time, the ratio of the other phases increases, and they become more visible in the XRD spectrum. In the trials performed with slag O2 it is particularly interesting to notice this phenomenon, because it provides information about what minerals are present when only a partial dissolution takes place. By comparing the results of the XRD analysis performed on the residues with the original slag composition and single mineral XRD patterns, almost all the peaks have been assigned to either merwinite, åkermanite or CaCl$_2$·nH$_2$O.

In Figure 10, the XRD spectra of both residues obtained after the neutralization with slag O2 are analyzed. Both spectra present the same phases, but the ratio differs as beforementioned. By considering only the intensity of the peaks, and comparing the results to each other, åkermanite (4CaO·1.5MgO·0.5Al$_2$O$_3$·3.5SiO$_2$, cyan circle mark) is the most abundant phase in both cases. From roughly 47% of the total sample in the 1 g trial it rises to 54% in the 2 g trial. Merwinite (3CaO·MgO·2SiO$_2$, yellow star mark) increases from 9% to 27%. Conversely, the CaCl$_2$·nH$_2$O content (magenta diamond mark) drop from 28% to 10% of the total sample. Therefore, this is evidence that contrary to the results for the other minerals, such as the ones present in slag sample O1, merwinite and åkermanite do not fully dissolve in the current experimental conditions. Given the low content % of merwinite in the 1 g trial and its initial % in the slag sample (~40%), merwinite seems to dissolve more than åkermanite. This is in line with previous findings [17], where a very different dissolution behaviors were found for the minerals tested and the pH levels of the solvent. Both merwinite and åkermanite show high dissolution levels and a fast dissolution rate at a pH value of 4. The dissolution rate at a pH value of 7 drops substantially, especially for åkermanite. However, it is still completely dissolved by the end of the trial. At a pH value of 10, the solubility drops significantly. Dicalcium silicate γ instead shows similar dissolution levels as merwinite and åkermanite only at a pH value of 4. Contrary to the other two, at a pH value of 7 the mineral dissolves with no visible difference compared to a pH value of 4. However, at a pH 10 the value of the dissolution level decreases. Although, it maintains a fast dissolution rate. In other studies, åkermanite is also shown to be less soluble, compared to other minerals like bredigite in a Tri-HCl solution of pH 7.4 [27].

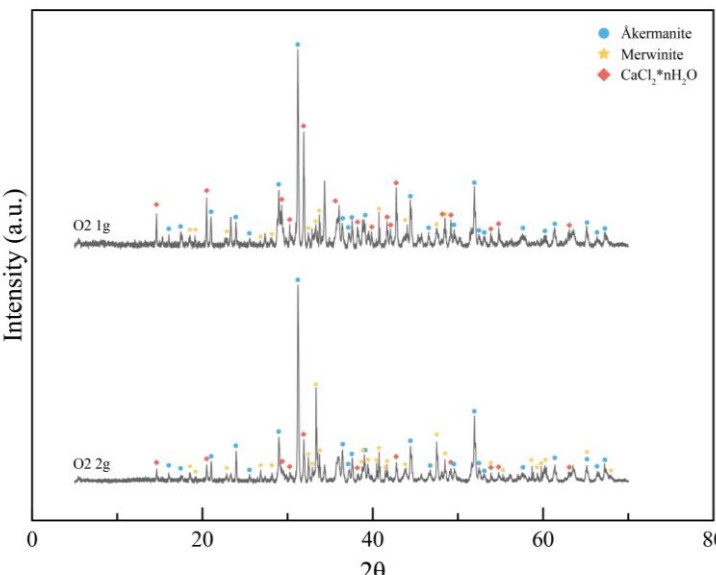

**Figure 10.** XRD spectra of the residues extracted after the neutralization trials of a 0.1 M HCl solution per-formed with 1 g and 2 g of slag O2. The intensities are normalized to a 100 to aid a proper comparison between the spectra. The peaks related with the phases åkermanite, merwinite and CaCl$_2$·nH$_2$O are highlighted by dots, stars and rhomboids at their respective peaks.

Nonetheless, the current study observes the dissolution behavior of minerals in a pH-variant environment, where the solubility of the species varies as well. Although, the results are compliant with previous findings made in environments using constant pH values. The lower dissolution kinetic of åkermanite compared to merwinite, especially around neutral values, means that more of the latter mineral is consumed when the pH level is crossing those values. This increases the pH value, until both minerals show very low solubilities. The faster kinetics of merwinite stifles the dissolution of åkermanite, by raising the pH value too fast for the second mineral to enable a complete dissolution.

### 4.3. Dissolution of Slag S1

Slag S1 (landfill slag) is the second-best performing material, despite the fact that it ranks third with respects to the total Ca (27.3%) and Mg (7.0%) contents. The sample is constituted of several of the minerals present in slags O1 and O2, as well as MgO and $Ca(OH)_2$. The pH levels obtained when using slag S1 are comparable to the ones obtained when using slags O1 and O2. The pH increase rate is slower compared to slag O1, and the final pH level obtained is between the results of slag O1 and slag O2 (Figure 4). When 2 g of slag are used, the slag behaves identically to sample O1. When looking at the XRD spectrum of the residues obtained to those that used 1 g of slag. Specifically, the results are similar to those when using slag O1, where the most abundant phase present is $CaCl_2 \cdot nH_2O$. This means that most of the mineral phases that are present dissolve during the trial. However, since slag S1 is formed by six major phases, its behavior is hard to predict by only observing the composition. Dicalcium silicate γ and bredigite fully dissociates when using slag O1, whereas åkermanite only partially dissolves when using slag O2, due to its low dissolution kinetic. In fact, faint traces of åkermanite can be detected in the XRD spectrum of the residues obtained when using 1 g addition of slag S1. This confirm that the mineral does not fully dissolve during the trials. No traces of merwinite can be found in the XRD spectrum. Therefore, it is assumed that most of it dissolve during the trial. This is also in line with the behavior of slag O2, that shows very low levels of residual merwinite when the slag is not overdosed (1 g trial). Periclase, as lime, is also a very hydrophilic mineral. Therefore, its dissolution is considered to be complete.

### 4.4. Dissolution of Slag S2

Slag S2 has been constantly an outlier in all the experiments performed. Contrary to the results from all the other slag samples, 1 g of slag is not enough to neutralize the acid content of the solution. This is quite surprising, since the slags have high concentrations of Ca (32%) and Mg (~3.8%). Moreover, dicalcium silicate γ makes up roughly 50% of the slag composition. This mineral is also abundant in slag type O1 (46.3%) and a S1 (19.9%). In addition, just like dicalcium silicate γ, mayenite ($12CaO \cdot 7Al_2O_3$) has been reported to have good hydraulic properties [28,29] and high solubilities even at high pH values [4]. Therefore, it is expected that the slag should show similar properties, if not better ones, than the previous slags. Instead, only when 2 g of slag are used, the material performs similarly to the other slag samples such as O1 and S1. The final pH value obtained when using a 2 g addition of slag S2 is approximately around 11, whereas additions of 2 g of the other slags could only rise the pH values between 8 and 9.6. Also, the final pH level is not reached gradually, but through a sharp increase around the end of the trial. This phenomenon happened also in the replication trial using the same quantity. Slag S2 is the only one containing mayenite, which is a mineral rich in Al. Al is the second most abundant element in slag sample S2, around 12% in weight, compared to a 2–3% content present in the other slags. As mentioned beforehand, Al is believed to counteract the rise of the pH value provided by elements such as Ca and Mg because its stable phases produce protons when being dissolved. This could explain why the pH values are so low compared to the ones obtained when the same weight of a different slag sample is used. The sharp rise in the pH value also is an interesting phenomenon worth exploring. According to the Purbaix diagram calculated in previous studies [21], after pH value 9 is reached, $Al(OH)_3$

should further hydrate, which would generate a proton. In theory this should decrease the pH value, but the opposite happens. No explanation of this phenomenon could be found, but it is believed to be connected to the phase transition of the Al species.

The XRD patterns of the residues obtained with slag S2 are completely different from the ones obtained when using the other three slags. Contrary to slag O2, that partially retained some mineral phases from the original sample, no traces of mayenite or dicalcium silicate γ can be found. However, contrary to slags O1 and S1, there is no formation of $CaCl_2·nH_2O$. Therefore, SEM-EDS imaging was used to understand the composition of the residues. The analyses show an abundant presence of small particles with sizes < 1 μm (points 8,7,10,12 in Figure 11). Given the very small size, these particles are considered to be precipitated compounds during the evaporation of the liquid phase. In fact, the slag samples have all been sieved to reach a size range of 25–50 μm. Thus, the presence of such a high number of very fine particles is otherwise unjustified. The fine particles contain Ca, O, Al and Cl as the main constituents. This explains why there is no trace of $CaCl_2·nH_2O$: the presence of dissolved Al bounds with Ca, Cl and O during the precipitation. Also, no traces of mayenite suggests that the mineral is the original mineral structure is completely dissolved and all the aluminum is dissolved in the acid.

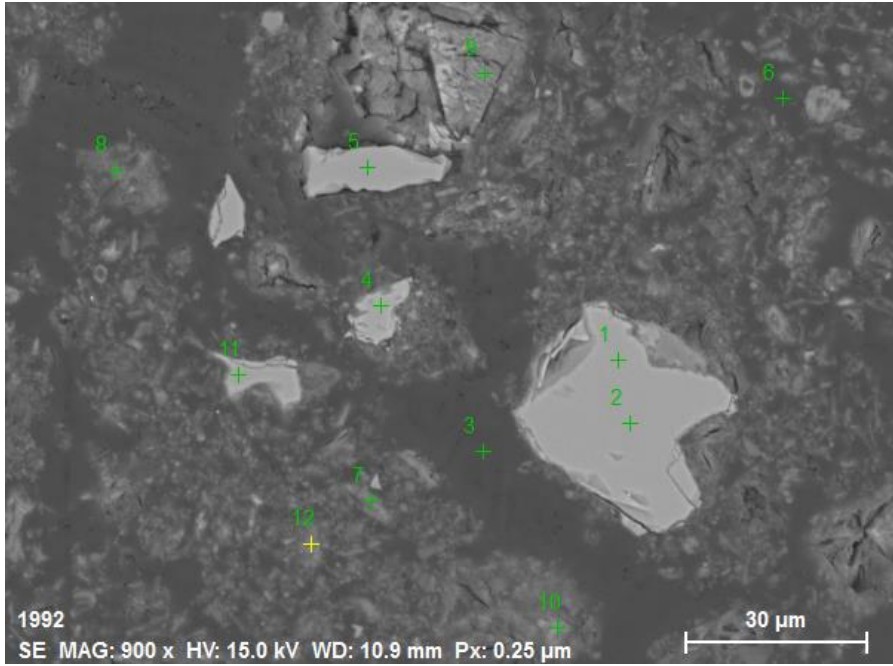

**Figure 11.** SEM-EDS image of the residues extracted after the neutralization trials performed with slag sample S2. Point analysis was used to determine the composition of the different phases present.

## 5. Conclusions

This study aimed at analyzing the effect of composition of different stainless-steel slags, regarding their ability to increase the pH values of standard acidic solution. The same slag samples used in previous studies conducted by the same authors are also used in the current publication. Although, the methodology developed in the current study is different than the one used in previous experimental trials. The current study used a fixed amount of slag additions (0.5, 1 and 2 g), and track the different pH values obtained, whereas previous experiments focused on reaching the same pH value, with a ranging weight based on the different slags used. Furthermore, industrial acidic waste waters have been replaced with 0.1 M HCl/$HNO_3$ solutions, and the quantity of solution to neutralize has been reduced from 1 L to 100 mL. Finally, the reacted slags and their leachates were extracted by drying the neutralized solutions. The residues were analyzed with XRD and SEM spectroscopies to determine their composition and infer the solubility of the original slag minerals.

The tested slags samples successfully rose the pH values to basic values of the 0.1 M monoprotic acid solutions (HCl or $HNO_3$) prepared in laboratory trials. The results of this study further strengthen the case for the use of stainless-steel slags for waste waters treatment. However, compared to commonly used materials such CaO, all the slags tested dissolved with a slower rate, showed a lower solubility and reached lower final pH values. Also, slag requires roughly 3 times the weight of lime to reach similar pH values. Nonetheless, for processes characterized by high retention times and where a high basicity is not required, Al-rich slags can be successfully be employed.

When the weight and particle size of the slag sample, as well as the acidic environment are controlled, the composition of the slag is shown to heavily influence their performance as reactants. Although, the bulk chemical composition by itself is not able to precisely predict the performance of the material by itself. Similar percentages of the most abundant elements in the slag samples often lead to different results when analyzing the obtained pH values. In these cases, the mineralogical compositions of slags are more important parameters to control. Specifically, the different solubilities of the mineral phases in water media are the parameters controlling the choices of the slag for such use. Minerals with good hydraulic properties such as dicalcium silicate γ or bredigite are favored for such applications. At the same time, the same minerals are often problematic for other applications, because of their tendency to leach. On the contrary, minerals such as åkermanite and merwinite are less soluble, which impact the final pH levels obtained by unit of weight.

This study also found that high percentages of Al are detrimental to a successful neutralization as they generate hydroxides that dissociate in free protons, which decreases the pH value of the solution. As a result, the same amount of Al-rich slag (S2), that was successfully used for the other samples to rise the pH level, was not sufficient to neutralize the acidic solutions. However, when the quantity of slag is adjusted accordingly, the final pH value is higher (11.0) than the one obtained with the other slags (8.2–9.6). Therefore, when high basicity is needed, rather than weight efficiency, Al slags can be successfully employed for the neutralization of wastewaters.

**Author Contributions:** Conceptualization, M.D.C.; methodology, M.D.C., R.K. and A.K. (Andreas Karlsson); formal analysis, M.D.C., R.K. and A.K. (Andreas Karlsson); investigation, M.D.C.; resources, M.D.C.; data curation, M.D.C. and R.K.; writing—original draft preparation, M.D.C.; writing—review and editing, A.K. (Andrey Karasev) and P.G.J.; visualization, M.D.C. and R.K.; supervision, A.K. (Andrey Karasev) and P.G.J.; project administration, P.G.J.; funding acquisition, P.G.J. All authors have read and agreed to the published version of the manuscript.

**Funding:** This research received no external funding.

**Institutional Review Board Statement:** Not applicable.

**Informed Consent Statement:** Not applicable.

**Data Availability Statement:** No new data were created or analyzed in this study. Data sharing is not applicable to this article.

**Conflicts of Interest:** The authors declare no conflict of interest. The funders had no role in the design of the study; in the collection, analyses, or interpretation of data; in the writing of the manuscript, or in the decision to publish the results.

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
