# Peer review of "Study of the Dissolution of Stainless-Steel Slag Minerals in Different Acid Environments to Promote Their Use for the Treatment of Acidic Wastewaters"

_applsci, doi:10.3390/app112412106_

Round 1

Reviewer 1 Report

            The research part is described correctly. I would suggest transferring some of the results to supplementary material. In the introduction, Fig. 1 should be deleted (reference suffices). The given reactions are based on the basic school knowledge. Please delete.

Reviewer 2 Report

The manuscript deals with analyzing the effect of the composition of different stainless-steel slags on their ability to increase the pH values of standard acidic solutions in order to promote their use for the treatment of acidic wastewaters. The topic is of practical interest.

The manuscript is well-written and logically presented. The conclusions are supported by the experimental data. The manuscript can be accepted to publication after minor revision. The remarks are listed below.

  1. Section 2.2 title, replace "Ph" with "pH".
  2. Line 151, abbreviation ICP-SFMS needs a full description at first mention.
  3. All data throughout the manuscript (Fig. 2-7, table 2) are presented as a single measurement result that is inappropriate. The corresponding coverage interval or SD values should be presented together with the average value.

Reviewer 3 Report

The paper entitled ‘’ study of the dissolution of stainless-steel slag minerals in different acid environments to promote their use for the treatment of acidic wastewaters’’ is an interesting approach with nice perspective for practical applications in the field of treatment of acidic wastewaters.

However, this article can thus be accepted in this journal for publication, although after a minor revision. The followings are my comments, which the authors may choose to address:

  1. The authors mentioned that they used SEM and EDS, I strongly recommended that SEM images should be added.
  2. in line 151, the authors mentioned that they tested HF, after that, the results of using HF were not mentioned in the discussion. This point needs to be clarified
  3. In Figure 8 and 9, I suggest if the authors use different color and the intensity values should be on y-axis
  4. The error should be corrected in line 462 and 554.
